# An Investigation of Efficiency Issues in a Low-Pressure Steam Turbine Using Neural Modelling

**DOI:** 10.3390/s24072056

**Published:** 2024-03-23

**Authors:** Marek Bělohoubek, Karel Liška, Zdeněk Kubín, Petr Polcar, Luboš Smolík, Pavel Polach

**Affiliations:** Research and Testing Institute Plzen, Tylova 1581/46, 301 00 Plzen, Czech Republic; belohoubek@vzuplzen.cz (M.B.); liska@vzuplzen.cz (K.L.); kubin@vzuplzen.cz (Z.K.); polcar@vzuplzen.cz (P.P.); polach@vzuplzen.cz (P.P.)

**Keywords:** condensing steam turbine, low-pressure turbine, machine learning, artificial intelligence, heat balance, nuclear power plant

## Abstract

This study utilizes neural networks to detect and locate thermal anomalies in low-pressure steam turbines, some of which experienced a drop in efficiency. Standard approaches relying on expert knowledge or statistical methods struggled to identify the anomalous steam line due to difficulty in capturing nonlinear and weak relations in the presence of linear and strong ones. In this research, some inputs that linearly relate to outputs have been intentionally neglected. The remaining inputs have been used to train shallow feedforward or long short-term memory neural networks using measured data. The resulting models have been analyzed by Shapley additive explanations, which can determine the impact of individual inputs or model features on outputs. This analysis identified unexpected relations between lines that should not be connected. Subsequently, during periodic plant shutdown, a leak was discovered in the indicated line.

## 1. Introduction

Machine learning (ML) is currently a discipline with very dynamic development due to the availability of computing power and the accumulation of vast amounts of data [1]. It involves a wide range of algorithms, approaches, and methods that use principles of empirical-based learning similar to human learning. Among these approaches, artificial neural networks (ANNs) are gaining popularity in solving complex problems [2].

Due to the versatility of machine learning approaches, which depend mainly on a sufficiently large volume of available data, the principles of neural modeling and artificial intelligence (AI) are gradually reaching more and more areas of human life. One of these areas is also the industrial environment, where neural modeling finds applications in controlling and regulating variables [3], anomaly detection [4], data classification [5], diagnostics [2,6], and dynamic system modeling or predictive maintenance [7,8]. Dynamic system models can be represented as a digital twin (DT) [4,9]—a digital counterpart of a physical system. The DT is often capable of processing real-time data, enabling smarter and more informed decision making when the system is not operating within the expected parameters. Hence, using the DT to simulate such conditions on the fly can reduce costs, increase efficiency, reduce vibration, or improve the long-term operability of the system [10,11].

The DTs are not necessarily based on ML—theoretical, phenomenological, or statistical models can represent many dynamic systems [9]. In some applications, such as aerospace or nuclear power, these models are preferred to those based on machine learning. Such practices are often adopted due to concerns regarding the unpredictable behavior of machine learning models in non-standard situations, and it is common in applications where conservative decision making is required for safety reasons. However, Hu et al. [2] suggest in a recent review that adopting ML models can improve the decisions of human operators in many areas. He also stressed that any model or algorithm employing machine learning depends on the input data quality and amount [2,12].

Many promising methods have been used in fossil fuel power plants. These methods are essential concerning nuclear power plants (NPPs) because secondary (non-nuclear) circuits share many similarities with fossil fuel power plants. Among others, Wisyaldin et al. [13] employed an ANN to predict the remaining useful life of circulating water pump bearings. In contrast, Ashraf et al. [14] employed an ANN with long short-term memory (LSTM) layers to identify which parameters influence the relative (shaft) vibration of a steam turbine. Dhini et al. [15] proposed a method to train networks that detect and classify faults in steam turbines.

Recent applications of ML are also focused on the field of nuclear energy. Sandhu et al. [16] review developments in condition assessment and AI applications of structural and mechanical systems in nuclear facilities. Tang et al. [17] analyze the intelligent demand scenarios in the whole industrial chain of the nuclear industry, investigate the research status of deep learning in the application fields corresponding to different data types in the nuclear industry, and discuss the limitations and unique challenges of deep learning in the nuclear industry. Huang et al. [18] briefly introduce modern AI algorithms such as machine learning, deep learning, and evolutionary computing and review several studies on the use of AI techniques for nuclear reactor design optimization as well as operation and maintenance. Li et al. [12] proposed a method that classifies transients during a nuclear reactor’s operation and is based on unsupervised machine learning.

The efficiency of the steam cycle can be measured by following the ASME PTC 6-2004 (R2014) code [19]. However, the measurement requires highly accurate and calibrated instrumentation and tightly controlled measurement procedures to yield results with low uncertainty [20]. Turbine manufacturers then use such results to calibrate a heat balance scheme (HBD) of the cycle [21,22]. However, the operator relies on data collected by a condition monitoring system, which are usually measured on a limited number of lines and are less accurate than dedicated measurements that follow ASME PTC 6-2004 [19,20,21]. These data can be compared with the calibrated HBD or used to compute the efficiency of individual components if the turbine manufacturer provides models or equations [21,22]. This research introduces a method that detects and localizes thermal anomalies in condensing low-pressure steam turbines without knowing such models or equations. The method requires training an ML or NN model that predicts steam parameters at the outlet from the low-pressure turbine and condenser backpressure from parameters measured in the path of steam flow through the turbine. Inputs that determine the condenser backpressure, such as the temperature and flow rate of cooling water through the main condenser, were omitted intentionally. The resulting models were analyzed using Shapley additive explanations (SHAPs) to determine the sensitivity of the outputs to inputs [23]. Such an analysis reveals relations between measured signals, including nonlinear and weak correlations. The individual correlations were assessed by experts who determined whether the correlation was expected. Unexpected correlations indicate that there is a fault in the system, such as a leak between lines, or the system is not operating within specifications.

## 2. Briefly on the Steam Turbines in Power Plants

A steam turbine can be characterized as a heat blade rotary engine in which mechanical energy on the shaft is obtained by the expansion of water vapor in one or successively in several turbine stages.

The thermal energy contained in the steam is converted to mechanical energy by expansion through the turbine. The expansion occurs through a series of stationary blades (nozzles) arranged within the turbine casing that orient the steam flow into rotary wheels attached to the turbine shaft. There, the steam flow exerts pressure on individual rotary blades, which have curved surfaces, allowing the formation of forces that cause the turbine to rotate. Each row of fixed nozzles and rotary blades is called a stage. In all turbines, the rotating blade velocity is proportional to the steam velocity passing over the blade. The typical main turbine in NPPs, in which steam expands from pressures around 6 MPa to pressures around 0.008 MPa, operates at 3000 RPM for 50 Hz grids [24].

Most NPPs operate turbine generators (TGs) with several rotors coupled into a single shaft train. These rotors typically include one multi-stage high-pressure (HP) and several parallel multi-stage low-pressure (LP) turbine rotors, a main generator, and an excitation system. Each turbine rotor is typically supported on two journal bearings, although some manufacturers also employ a single-bearing configuration for LP turbine rotors.

## 3. Description of a Problem to Be Solved

The specific problem that we solved was related to the issue of the efficiency of steam turbines and the related decrease in the generated electric power in a WWER-type of NPP. This power plant consists of two similar units, each containing a 1000 MW TG, whose diagram of the thermal cycle is shown in Figure 1. Both TGs consist of one HP turbine and three LP turbines, and the topologies of both steam turbines are the same. The HP turbine is supported by two tilt-pad journal and one tilt-pad thrust bearings, each LP turbine rotor is supported by a single tilt-pad journal bearing located behind each rotor (a standard direction to describe the shaft train is assumed, i.e., the HP rotor is located at the front of the train and the generator rotor is located at the end of the train), and the generator is supported by two tilt-pad journal bearings.

After experiencing blade flutter, the LP turbines of Units 1 and 2 were reconstructed in 2014 and 2015, respectively [25]. The reconstruction brought the rated electric power of each unit up to 1086 MW. After the reconstruction, the operator identified several issues: elevated backpressure in main condenser 3 of Unit 1 leading to reduced thermal efficiency, vibration problems due to rotor–stator rubbing, and suspected thermal deformation of turbine foundation. The issues occurred more often in Unit 1 than in Unit 2. Standard approaches, such as a correlation analysis considering more than 300 operational parameters, did not provide any definitive answers or solutions.

In 2020, a thorough investigation of the thermal efficiency issue was launched using ML methods. First, historical process data such as fluid temperatures, pressures, and flows were collected from the central storage of technological data (CSTD). The starting point of the evaluation was to compare historical data with heat balance diagrams (HBDs). The HBD contains a schematic representation of the steam cycle shown in Figure 1 together with expected temperatures, pressure, heat flows, and enthalpies. This evaluation used a simple convolutional neural network and confirmed anomalies in the LP turbines of Unit 1, with a prominent anomaly in LP3 where increased backpressure was observed during a nominal-load operation. The second step was to tune AI models to compare both turbines and all six LP turbines. For this purpose, 24 of the available 66 process parameters were selected to play a significant role in anomaly detection. Finally, the AI model was used to compare and explain the anomalies of all six LP turbines, using the developed sensitivity and predictor power analysis of the trained ANN. The analysis indicated that the sealing area of the steam circuits leading to the LP heater was suspected. The last step was to perform experimental validation to verify the assumptions given by the ANN outputs. During the test, the backpressure changed, with later inspections confirming faulty components causing the leaks.

After 3 years, almost all spare problems have been solved, but the strange behavior still exists. This paper presents an AI-driven strategy to find the strange behavior of the first turbine generator (TG1) based on steam parameters.

The two most appropriate artificial neural network approaches were chosen to solve the problem: the feedforward neural network (FNN) and the LSTM. These approaches were, e.g., successfully used to predict bearing vibrations: FNN in [14] and LSTM in [13]. Data time histories of operating parameters (temperatures, pressures, water level, and power output) were taken from the CSTD.

## 4. Methodology

As it was already mentioned, two deep learning approaches were used to investigate efficiency issues in LP steam turbines: the FNN and the LSTM network.

### 4.1. Feedforward Neural Network

The FNN consists of serially arranged fully connected layers, within which neurons are connected in the sense of “every with every” [26]. While the information may pass through multiple hidden nodes, it always moves in one direction [27]. This means that the connections between nodes do not form any cycle. Contrary to the FNNs, recurrent neural networks (RNNs) allow information to cycle through certain pathways.

The unidirectional propagation of the information in the FNN from the input layer through the hidden layers and to the output layer is depicted in Figure 2. Input layer $l_{0}$ passes inputs stored in vector $x=\left(x_{1},\;\ldots ,\;x_{m}\right)$ to the first hidden layer $l_{1}$, if any hidden layer exists. Next, the nodes in the hidden layers obtain outputs from the previous layer, transform them, and send them to a subsequent layer. Finally, data are collected in the output layer $l_{H+1}$, which arranges and produces the output stored in vector $y=\left(y_{1},\;\ldots ,\;y_{n}\right)$ [28].

The data transformation at the level of a single node can be mathematically described as [27]
(1)$$h_{i}^{k}=b_{i}^{k}+\sum_{j=1}^{r_{k-1}}w_{ij}^{k}o_{j}^{k-1},i=1,\ldots ,r_{k},$$
where $o_{i}^{k-1}$ is the output from node *i* in layer $l_{k-1}$, $r_{k-1}$ is the number of nodes in layer $l_{k-1},\;w_{ij}^{k}$ is the weight for node *i* in layer $l_{k}$ for data incoming from node *j* in layer $l_{k-1}$, and $b_{i}^{k}$ is the bias for node *i* in layer $l_{k}$. Weights $w_{ij}^{k}$ and biases $b_{i}^{k}$ are independent parameters whose values are found through optimization, also referred to as training. Value $h_{i}^{k}$ is further transformed by transfer function *g* with the general prescription
(2)$$o_{i}^{k}=g\left(h_{i}^{k}\right).$$

The most widely used types of transfer functions are available, e.g., in [29].

The final values of independent parameters and their total number are influenced by initial conditions and so-called hyperparameters, which define the ANN structure and training process, and are described in [28,30].

### 4.2. Long Short-Term Memory Network

In the case of a time-series processing problem (in general, a sequence-to-sequence task), the FNNs described above allow for representing the dependence of the output quantity $y_{t}$ only on the state of the system in predefined time steps, in the sense of $y_{t}=f(x_{t},x_{t-1},x_{t-2},\ldots ,x_{t-m})$, where $f\left(.\right)$ represents a general nonlinear transformation. Alternatively, the states correspond to explicitly defined (non)linear combination.

If the actual state of the system is dependent on a previously unknown number of previous time steps or if this dependency is variable over time, the above description may be insufficient. Recurrent neural networks (RNNs) of the LSTM (long short-term memory) type can successfully eliminate this deficiency [31]. These contain a so-called hidden state $h_{t}$ representing the long-term dependencies contained in the system. LSTMs are similar to standard RNNs (see, e.g., Elman [32]), but in this case, each node is replaced by a so-called memory cell, which, by using a system of gates, allows the long-term state of the system to be preserved without vanishing or exploding of the gradient of the backpropagation error [33]. Specifically, we are talking about three gates—input, forget, and output gates—which decide what information is added to, removed from, or removed from the memory cell. This mechanism then allows for the selective storage of key information, and conversely the suppression of less relevant information from the long-term history of the system [31,34].

The mathematical implementation of the above mechanism is illustrated in Figure 3, where the structure of the LSTM memory cell is shown. The LSTM cell operates with three types of inputs, namely the hidden state $h_{t-1}$, the cell state $c_{t-1}$, and the inputs $x_{t}$. In each time step, a new cell state $C_{t}$ is defined using relations (3)–(6) representing the forget gate (output vector $f_{t}$) and input gate ($i_{t}$ and $c_{t-1}$). The new value of the hidden state $h_{t}$ is then given by Equations (7) and (8) and is thus a (nonlinear) function of the state of cell $c_{t}$, the inputs $x_{t}$, and the previous value of the hidden state $h_{t-1}$ [35]:(3)$$f_{t}=\sigma \left(W_{xh,f}x_{t}+W_{hh,f}h_{t-1}+b_{f}\right),$$
(4)$${\tilde{c}}_{t}=tanh\left(W_{xh,c}x_{t}+W_{hh,c}h_{t-1}+b_{c}\right),$$
(5)$$i_{t}=\sigma \left(W_{xh,i}x_{t}+W_{hh,i}h_{t-1}+b_{i}\right),$$
(6)$$c_{t}=\sigma \left(f_{t}⊙c_{t-1}+i_{t}⊙{\tilde{c}}_{t}\right),$$
(7)$$o_{t}=\sigma \left(W_{xh,o}x_{t}+W_{hh,o}h_{t-1}+b_{o}\right),$$
(8)$$h_{t}=tanh\left(c_{t}\right){⊙o}_{t},$$
where tanh is the nonlinear activation function; the operator ⊙ represents the element-wise product; $W_{xh,f},W_{hh,f},W_{xh,i},W_{hh,i},W_{xh,o},W_{hh,o},W_{xh,c}$**,** and $W_{hh,c}$ are the weight matrices for individual gates; and $b_{f},b_{i},b_{o}$**,** and $b_{c}$ are bias terms for individual gates.

### 4.3. Interpretation of NN Models

From the point of view of mathematical description, NN prediction is (with exceptions) a strongly nonlinear transformation of input values, and the input–output characteristics of NN values may not be fully explicit. Given this fact, several methods have been developed over the years to deal with at least a partial interpretation of NN outputs [36,37,38]. Many of these methods are based on introducing small (or large) disturbances at the input and subtracting their influence from the output of the network. The most typical output of these methods is the classification of the power of predictors representing the contribution of each input to the final decision of the neural network.

For the case of this work, a methodology based on the definition of Shapley values and belonging to the SHAP family of methods has been proposed, which are currently among the most popular methods used to interpret ML models and their decisions [23], next to lime (local interpretable model-agnostic explanations [39]) or CAM (class activation maps [40]) approaches.

SHAP-type methods are based on the definition of Shapley values defined in the context of game theory, where these values reflect the player’s contribution to a cooperative game. Unlike other estimation approaches, the Shapley method calculates the player’s contribution to themselves and in the context of cooperation with other players. In the context of the ML area, the principle of the Shapley method is used to determine the significance of a change in input variables on an output variable, i.e., to define the so-called power of predictors. This is achieved by summing and specifically averaging the differences in the contributions of each input concerning the average prediction [37]. Specifically, for a set of discrete inputs, the Shapley value φ of the ith input can be defined as
(9)$$\varphi_{i}\left(x_{\emptyset }\right)=\sum_{n=1}^{N}w_{n}\sum_{l=1}^{L}\Delta f_{l}\left(x_{\emptyset }\right),\;\;\Delta f_{l}\left(x_{\emptyset }\right)=f\left(x_{l}\right)-f\left(x_{\emptyset }\right),$$
where N is the number of input variables, $\Delta f_{l}\left(x_{\emptyset }\right)$ corresponds to the difference between the average output of the model and the output also containing the lth unordered tuple (i.e., combination) of modified input variables. This combination always consists of n inputs and is multiplied by a weighting coefficient $w_{n}$, which provides a balance between the unequally represented number of combinations L. Specifically, for these variables,
(10)$$w_{n}=\frac{1}{n\left(\begin{array}{c} N\\ n \end{array}\right)}=\frac{n!\left(N-n\right)!}{nN!},\;\;L=\left(\begin{array}{c} N\\ n \end{array}\right)=\frac{N!}{n!\left(N-n\right)!}.$$

Since the problem operates with a regression model containing continuous input and output variables, the above calculation of Shapley values was modified accordingly. To eliminate order differences between the input variables, we worked with normalized x inputs (using z-score), for which n elements were always modified using the perturbation $\xi_{j}$. Specifically, the following was used:(11)$$\varphi_{ij}\left(x_{\emptyset },\xi_{j}\right)=\sum_{n=1}^{\tilde{N}}w_{n}\sum_{l=1}^{L=\left(\begin{array}{c} N\\ n \end{array}\right)}\left|\Delta f_{l}\left(x_{\emptyset },\xi_{j}\right)\right|,$$
where, for example, for the first perturbed tuple (l=1) of two perturbed variables (n=2), we can write
(12)$$x_{\emptyset }+\xi_{j}={\left[x_{1\emptyset }+\xi_{j},\;\;\;x_{2\emptyset }+\xi_{j},\;\;\;x_{3\emptyset },\;\;\;\cdots \;\;x_{N\emptyset }\right]}^{T}.$$

The size of the perturbations introduced can be set arbitrarily, resulting in a generally infinite number of test variants. From this, it is necessary to select J discrete values appropriately representing the working interval of the given variable. The total number of variants of model outputs to be implemented in this way to obtain predictor power would be $M=2^{N}J$, from which the computational cost of the whole problem can be further reduced by reducing the number of cooperative tuples ($N\to \tilde{N}$). For the full computation of the Shapley values, the number of groups of cooperative tuples generated is the same as the number of input values, i.e., N. However, for large values of N, this entails a very high number of combinations and hence excessive computational cost. Therefore, we reduce the number of groups of generated cooperation tuples to $\tilde{N}$.

The resulting contributions of the individual variables are then summed over all tested perturbation sizes and normalized using the maximum norm to avoid disproportionate differences between them. Thus, the resulting value of the power of the i-th predictor is calculated as
(13)$$\varphi_{i}\left(x_{\emptyset }\right)=\frac{1}{\underset{i}{\max}\{\tilde{\varphi_{𝚤}}\left(x_{\emptyset }\right)\}}\tilde{\varphi_{𝚤}}\left(x_{\emptyset }\right),\;\;\tilde{\varphi_{𝚤}}\left(x_{\emptyset }\right)=\sum_{j=1}^{J}\frac{\varphi_{ij}\left(x_{\emptyset },\xi_{j}\right)}{\underset{i}{\max}\left\{\varphi_{ij}\left(x_{\emptyset },\xi_{j}\right)\right\}}.$$

### 4.4. Input Data Preparation for Neural Networks

For the training of neural network models, a dataset comprising 66 diagnostic quantities collected over 3 years sampled at a period of one minute was provided by the plant staff. These data encompassed all six low-pressure (LP) parts of the turbine. Although the sampling period of one minute seems very long, the simulated dynamics are slow because it depends on outdoor temperature and grid demand for electricity predominately. The base periods of these two phenomena are on the order of a day–night cycle.

Through expert analysis, the dataset was reduced to 24 inputs, listed in Table 1, which were considered to have a potential impact on the outputs of each LP part, listed in Table 2. On the one hand, this reduction eliminated signals that showed a constant characteristic, long-term errors, or where the operator assumed very low accuracy of the recorded values. An equally significant proportion of the signals was then removed by the plant’s experts for being of low significance in terms of the physical operation of the turbine and the assumption that they were less likely to affect the overall efficiency decline of the turbine. Moreover, some signals with a known strong linear correlation to outputs, such as the temperature of cooling water, which determines the backpressure in the main condensers, were omitted deliberately. Specifically, the selected parameters mainly corresponded to the positions of steam valves, as well as pressure and temperature measured at various lines connected to the LP turbine.

To ensure a reliable reference point for analysis, the second low-pressure (LP2) part was specifically chosen as the reference for training data, because it had not shown any significant anomalies or irregularities. By utilizing LP2 as the reference, the neural network models were trained to understand the typical behavior and dynamics of the system. It provided a stable foundation for comparison and analysis between the LP parts of the turbine generator of Unit 1 (TG1) and the turbine generator of Unit 2 (TG2). As TG1 and TG2 operate under distinct operational conditions, training separate neural networks for each machine allowed us to capture their unique characteristics and behaviors.

To optimize the machine learning models, a preprocessing step was performed on the input data. This involved selecting a specific subset of diagnostic data collected during turbine operation when the power output exceeded 675 MW (out of a total approximate power output of 1100 MW). This power threshold was chosen to exclude highly specific data from turbine startup and shutdown periods, while still capturing the dynamics of events related to the problematic power transfer of 800 MW. The explanation and analysis of the problematic power transfer occurring around 800 MW can be found in Section 5.1 of this paper.

Filtering was then applied to the input data to remove outliers from the signals. These points could be expected to be subject to error in the measurement or recording device, which could cause undesirable distortion of the input data needed for the NN training process. The filtering itself was implemented using a moving median filter with a moving window, often referred to as a Hampel filter. The particular condition for outlier detection has the following form:(14)$$|x_{k}-median\left(X\right)|\geq c\cdot f\cdot MAD_{k},$$
(15)$$c=-\frac{1}{\sqrt{2}\Phi^{-1}(\frac{3}{2})}\approx 1.4826,\;\;\;MAD_{k}=median\left[\left|x_{k}-median\left(X\right)\right|\right],$$
where X represents the set of input samples $x_{i},\;i=1,2,\;\ldots ,\;N$ and $k=\left\lfloor \frac{N}{2}\right\rfloor$. $MAD_{k}$ refers to the median absolute deviation, *f* is the threshold factor, and Φ is the complementary error function [41]. In this case, f=5 and N=5000 were applied to filter the data. The outliers detected in this method were subsequently replaced by the value of the calculated local median. Furthermore, time samples with faulty or incomplete data were adjusted. The missing samples were linearly interpolated in cases when only a few samples were missing. In this case, when the missing chunk of the time series was longer, the missing chunk was synthetized by adding white noise to the interpolated values.

In the training process of neural networks, it is important to divide the available data into three distinct datasets: training, validation, and test datasets. That is why the prepared data for LP2 were divided into a 90% portion for training and validation (specifically 70% for the training set and 20% for validation), with the remaining 10% reserved for testing. The validation dataset was used to indicate that the best level of generalization of the neural network was reached (with the absence of underfitting or overfitting). In addition to the maximum number of training epochs, an early stopping condition was defined by the number of 10 consecutive training epochs during which the loss function over the validation set was not reduced. The test dataset was then used to independently evaluate the performance of the resulting trained model. In all cases, the validation and test datasets were chosen as one continuous time series.

Before commencing the learning process, the input data were normalized using the z-score [42] method. To train an LSTM neural network, the complete time series was divided into equally sized sections—time sequences—without any time overlap between them. The length of these sequences was one of the optimized hyperparameters of the LSTM network. Another optimized parameter was the size of minibatches corresponding to one learning batch/training iteration of the network. Thus, the resulting input of the LSTM network was a three-dimensional matrix with sizes corresponding to the number of input variables, the length of the time sequence, and the size of the minibatch. The overall process of data preparation and the workflow during task processing are depicted schematically in Figure 4.

## 5. Results and Discussion

The results of this study are presented in this section, which involves an analysis of the HBDs and training models for all three low-pressure parts of both turbines using the neural network architectures mentioned above: FNN and LSTM. Following the training process, we proceeded to compare the results obtained from the FNN and from the LSTM models and conducted a thorough comparative analysis. The entire task was solved in MATLAB software (versions R2021a, R2021b, and R2022a), including the actual training of the neural networks.

### 5.1. Heat Balance Diagrams

Firstly, our focus was on comparing the behavior of TG1 and TG2 of the power plant. The same configuration of TG1 and TG2 led us to compare them with a method based on the input/output model. At our disposal, we had HBDs for 14 different operation states.

These HBDs were used to select a starting set of measured parameters suspected to affect the steam parameters at the outlet from the turbine. In total, 66 temperatures, pressures, flow rates, and valve relative positions in steam lines and main condensers directly connected to the LP turbines were selected. First, the measured values were compared to HBD calculated values. This comparison did not provide any meaningful results, because many measured values differed from the theoretical ones that were introduced in HBDs. Therefore, the first regression shallow network was trained. The input parameters were the calculated values (temperatures, pressures, power output) and the output for this network was the number of current HBDs. This shallow neural network helped us to determine the closest theoretical operational state and the related HBD. The measured data from three one-year campaigns, which include power load and nominal states (full load), were used as input to this neural network. The output showed that TG1 and TG2 behave similarly up to 800 MW. From 800 MW, the number of predicted HBDs was different for TG1 and TG2. This was the first significant proof that the behavior of TG1 and TG2 is different. This behavior persists up to full load.

As the most significant value, the pressure drop on the last-stage blade (the third low-pressure part—LP3) and the pressure in the main condenser 3 part were marked. Figure 5 shows the dependency of pressure drop on the last-stage blade LP3 part on the turbine power output. At 800 MW, the linear relationship changes its slope. On the other LP parts of TG1, the slope change was also observed but it was always less intense. On TG2, this behavior was not observed.

Figure 6 shows pressure differences in individual main condensers of TG1. It is obvious that the pressure is up to 2 kPa higher in main condenser 3 during nominal operation, compared to the other main condensers.

Another indicator that something is wrong in the LP3 part was the behavior of the water level in the condenser hotwell. All three main condensers are connected, so the water level is based on the pressure. In addition, the pressure in the main condenser depends on the cooling water temperature. Figure 7 shows the dependency on water level in the main condensers in TG1 and TG2. The color represents the temperature of cooling water. It can be seen that there is a linear dependency of pressure on the temperature of the cooling water and there are anomalies in main condenser 3 of TG1 where the water level is changing, which is not true for main condenser 1. Also, this is not occurring to TG2. This behavior led us to the idea that the pressure in main condenser 3 of TG1 behaves independently and causes decreasing efficiency. The root cause analysis was performed afterward by training the LSTM and FNN neural networks of LP parts.

### 5.2. Architecture and Results of Trained Neural Networks

In terms of mathematical–physical analysis, a turbine is a complex heavily nonlinear dynamic system whose behavior is dictated by many input parameters with variable time delays. As outlined above, the purpose of this work was to develop a machine learning model capable of using preidentified input parameters to predict four pressure- and temperature-related output quantities for relevant LP parts to analyze quantitative and qualitative differences across the entire TG.

From the machine learning perspective, this type of task is a model representing a system with memory—supervised machine learning (SML)—and a regression type of output. Concerning the complexity of the analyzed system and the need to capture the inertia of the system, two different approaches based on neural networks, as was already stated, were chosen, LSTM and FNN with a time delay, and a time delay neural network (TDNN).

For both approaches, a set of hyperparameters was carefully selected, and the grid search method was employed to identify the optimal combination of parameters that best addressed the problem at hand. The hyperparameter optimization process was iterative, initially conducted on a shorter time series to expedite the training phase. NN training for each combination of input hyperparameters was performed on an untrained network with an initial random initialization of the weights and bias values with a normal distribution around zero, which was generated independently for all network weights. The loss function curve had the expected exponential shape for most of the chosen combinations of hyperparameters. It is important to note that while the final values of the hyperparameters may not represent the absolute best combination, they proved to be sufficiently accurate for the purposes of prediction, as demonstrated in Figure 8. Figure 8 presents a comparison of the network trained using optimized final values of hyperparameters from Table 3 and Table 4. Each graph is accompanied by the corresponding root-mean-square error (RMSE), which expresses the error between the predicted and measured data. The low error observed between the predicted and measured data signifies that the neural networks have effectively learned the underlying patterns and relationships within the system. It also shows that neural models have a sufficient degree of generalization in the absence of overfitting of the model. This is demonstrated by good prediction on both the training dataset (before the dashed line) and the test dataset (after the dashed line) in Figure 8.

As mentioned above, in the first phase of the solved task, the last 10% of the data served as a test dataset. The overall course of the optimization process mainly included experiments in the field of network architecture, number of neurons, number of learning epochs, learning rate, training algorithm, size of minibatch, or size of traffic delay, using the grid search technique. The evaluation criteria were always metrics—root-mean-square error (RMSE), mean square error (MSE), and values of the loss function over the validation data, which appropriately penalizes predictions with a greater deviation from real data. The total number of hyperparameter combinations tested was greater than 250 [43].

Once the predictive capabilities of the neural network models were demonstrated, it was possible to proceed to the next stage of problem solving. This consisted of training over the full set of available data for the LP2 part domain, comparing them qualitatively and quantitatively with the first low-pressure (LP1) and LP3 parts. Given the physical differences (as it was declared in Section 5.1), each of the turbine generator models was trained and compared separately.

The results of the comparative analysis confirmed the initial assumptions, i.e., the physical mismatch in behavior of the LP3 part manifested in the different values of the condenser backpressure. This is well illustrated by the outputs of both neural networks (see Figure 9). For the training set TG1-LP2, we obtained a good agreement with the first LP part, but in the last LP part, a constant difference (offset) between predicted and measured values was visible. Assuming proper training of the neural models, these results can then be interpreted as confirming the assumption of different physical behavior between LP1/LP2 parts and the last LP3 part.

### 5.3. Interpretation of Models and Comparative Analysis across LP Parts

Both types of trained models (FNN and LSTM) confirmed the assumptions of the different behavior of the TG1-LP3. The question that the trained models were to answer was therefore primarily the reason and origin of this different behavior.

As mentioned in Section 4.3, NNs generally represent a strongly nonlinear input-output data transformation, and their interpretation may not be entirely explicit. First, classical sensitivity analysis was used to identify the input variables that influence the different behavior of the TG1-LP3 part. This involved introducing a 10% perturbation of the values of the individual input variables (in a positive or negative direction, depending on whether there was a correlation between the variables corresponding to a positive or negative Pearson correlation coefficient value) and monitoring the shift in the response on the actual TG1-LP3 data. Specifically, it was then monitored for the elimination of the offset and a general reduction in the resulting deviation between the predicted and real data, in terms of the RMSE metric. If the initial estimate of the input 10% change was too coarse, refinements were made in 1% steps. The set of variables with the lowest RMSE values were found to be suspected of generating anomalies.

A representative result of this initial analysis (specifically for the variable “steam at 3. extraction from LP3, validated”) can be seen in Figure 10, where, after adjusting the input variable, the predicted and real values for a given TG1-LP3 are significantly similar.

The results of this sensitivity analysis were followed up by defining the overall power of the predictors, in the sense of the SHAP method, which considers, in addition to the partial contribution of one input variable, the importance of the variable in cooperation with the others, as described in Section 4.3. In the case of this analysis, the model parameters chosen were J=6 and $\;N=24\to \tilde{N}=3$.

There was a very good agreement between the LSTM- and FNN-type models for the variables/predictors extracted, and agreement was also observed with the results of the previous sensitivity analysis (compare Figure 10 and Figure 11). Five input variables that, according to the NN models, influence the backpressure in main condenser 2 and can cause pressure anomalies there are presented in Table 5.

The results of the NN models were presented to experts for final review. The experts assessed whether the relations found by the NN models were expected and possible. The high response of backpressure in the main condenser to the steam pressure at 3. extraction from LP was unexpected because there was no direct line between the main condenser and this extraction line (see Figure 1). Each extraction line is fitted with a drain connection that leads to the hotwell and condenser shell through an expander. Theoretically, the expander should stop steam from flowing to the main condenser if any leak occurs in the drain connections. However, such a leak was the only plausible hypothesis. Other high responses indicated possible problems within a gland seal system.

In reaction to the results, the operator decided to change the pressure in the steam seals to 125% of the nominal value and to close some valves fitted on by-passes connecting the sealing steam lines with the LP regeneration system (LPH1–LPH4). In addition, the visual inspection of the drain connections and their outlets to the main condenser was scheduled for the next scheduled outage of Unit 1. Although the inspection was challenging because the inspected lines were located inside the main condenser and difficult to access, the inspecting person found holes in the fittings of two drain connections in main condenser 3.

## 6. Conclusions

This paper has introduced machine learning methods that can help in detecting thermal anomalies in the steam cycle. The use of these methods was demonstrated in the example of six condensing low-pressure steam turbines installed in a nuclear power plant.

The first method is capable of finding a heat balance diagram that is closest to the measured data. This method can help personnel in comparing actual data with theoretical data more easily, and because of its simplicity, it can even be used in real-time in condition monitoring applications.

The second method uses neural network models to help detect and localize thermal anomalies. These models are trained using measured inputs and outputs. However, the inputs with a known strong linear relation to any output are omitted deliberately so that the models can better target weak and nonlinear relations. The dependence of outputs on inputs was evaluated by employing Shapley additive explanations and perturbation analysis. Experts reviewed the results and identified unexpected correlations that can indicate faults. In the case of a diagnosed system, the expert evaluation of unexpected correlations led to adjustments such as increasing steam seal pressure and closing valves in the LP regeneration system. Moreover, the visual inspection during a scheduled outage confirmed leaks in the main condenser drain connections.

In summary, this study shows the possible application of neural network models that are trained without knowledge of certain inputs in condition monitoring and diagnostics of thermodynamic systems. The authors believe that a similar approach can be used to detect existing and latent anomalies in other scenarios, thereby bolstering the efficiency, safety, and reliability of critical infrastructure, including nuclear power plants.

## Figures and Tables

**Figure 1 sensors-24-02056-f001:**
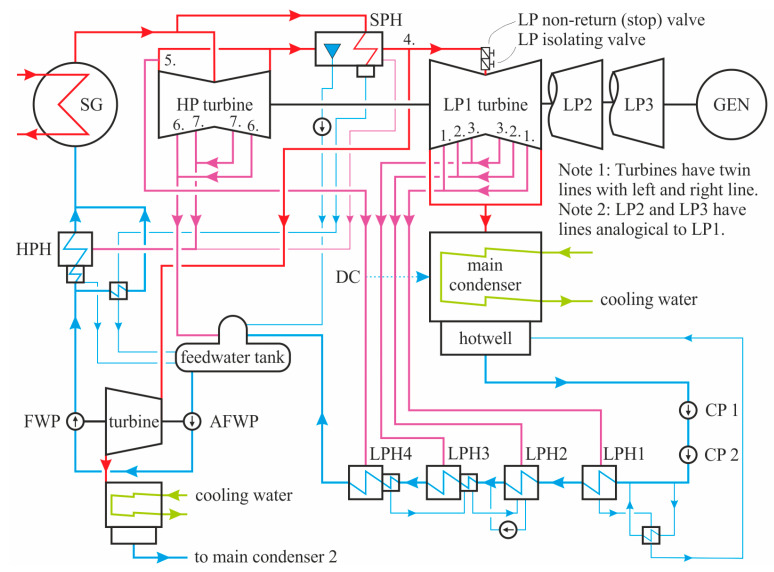
Simplified diagram of the thermal cycle of a monitored 1000 MW unit. SG, steam generator; HP, high pressure; LP, low pressure; 1.–7., extractions; GEN, generator; DC, drain connections; HPH, high-pressure heater; LPH, low-pressure heater; SPH, steam separator and heater; FWP, feedwater pump; AFWP, auxiliary feedwater pump; CP, condensate pump.

**Figure 2 sensors-24-02056-f002:**
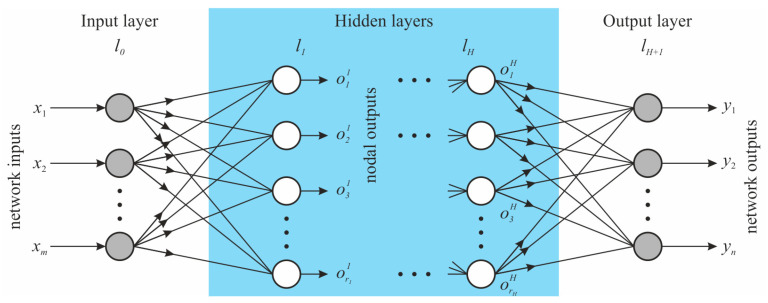
Scheme of a feedforward neural network with *m* inputs, *n* outputs, and *H* hidden layers.

**Figure 3 sensors-24-02056-f003:**
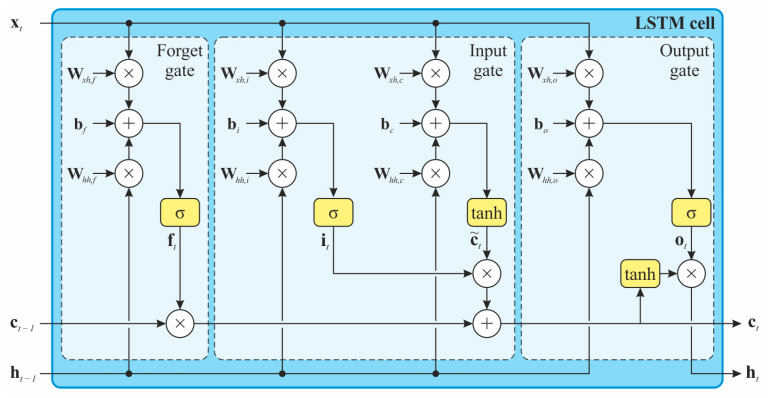
Structure of an LSTM cell with forget, input, and output gates, and two states—cell state $c_{t}$ and hidden state $h_{t}$—which is also the cell output, $y_{t}$.

**Figure 4 sensors-24-02056-f004:**
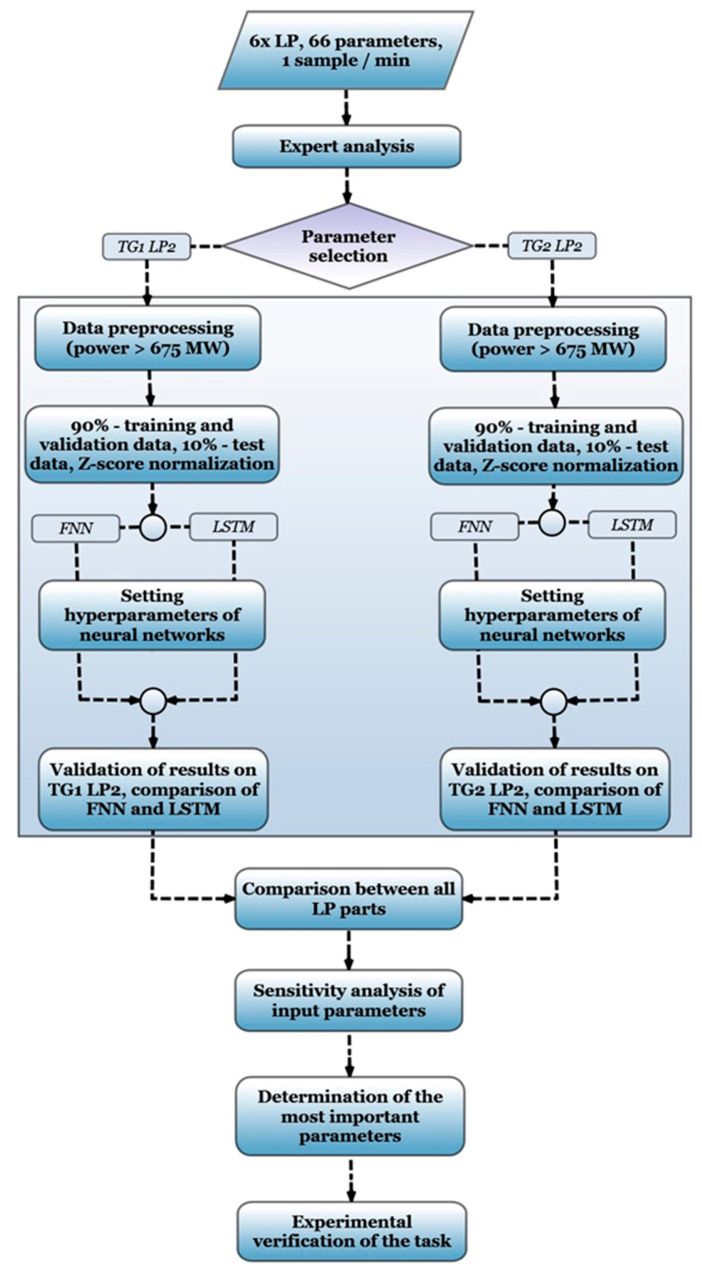
Flowchart of the workflow of the analyzed task.

**Figure 5 sensors-24-02056-f005:**
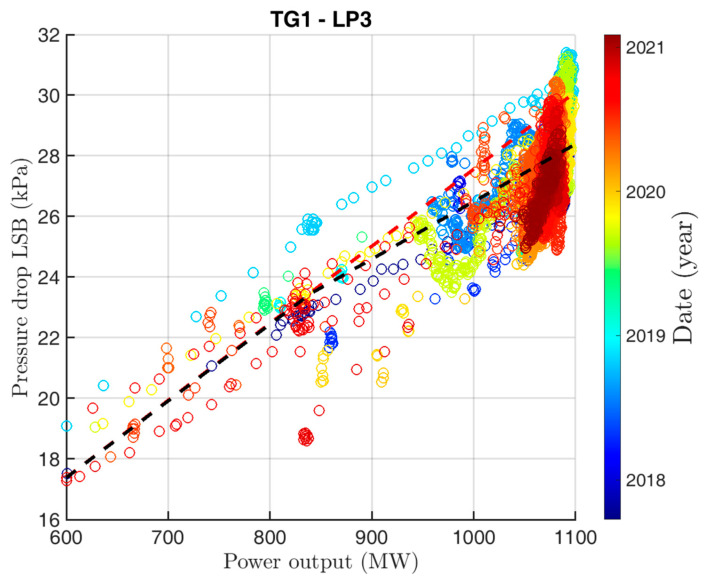
Dependence of the pressure drop on the blade of the last stage (3rd LP part) on the turbine power. The dashed lines in the graph indicate the difference between the expected slope (red) and the real slope (black) of the pressure drop.

**Figure 6 sensors-24-02056-f006:**
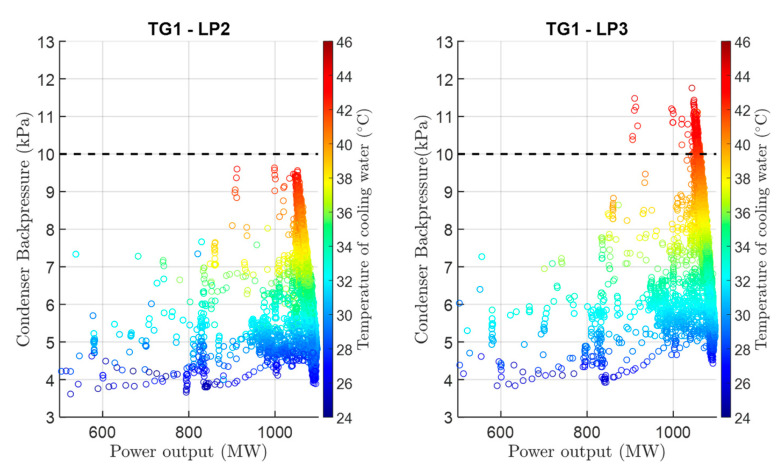
Condenser backpressure dependence on turbine power for reference TG1-LP2 and analyzed TG1-LP3 with a clearly visible difference in the maximum condenser backpressure value of about 2 kPa.

**Figure 7 sensors-24-02056-f007:**
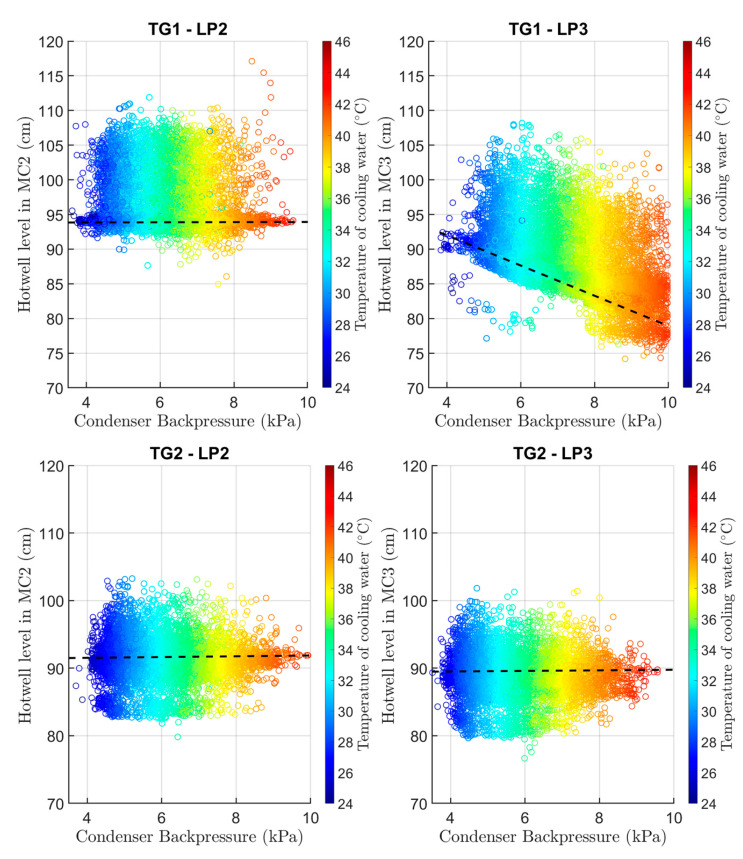
Dependence of hotwell level in the main condensers (MCs) on backpressure for TG1-LP2 (top left), TG1-LP3 (top right), TG2-LP2 (**bottom left**), and TG2-LP3 (**bottom right**) with anomalous behavior for TG1-LP3, showing the undesirable linear dependence of hotwell level on backpressure and temperature of the cooling water.

**Figure 8 sensors-24-02056-f008:**
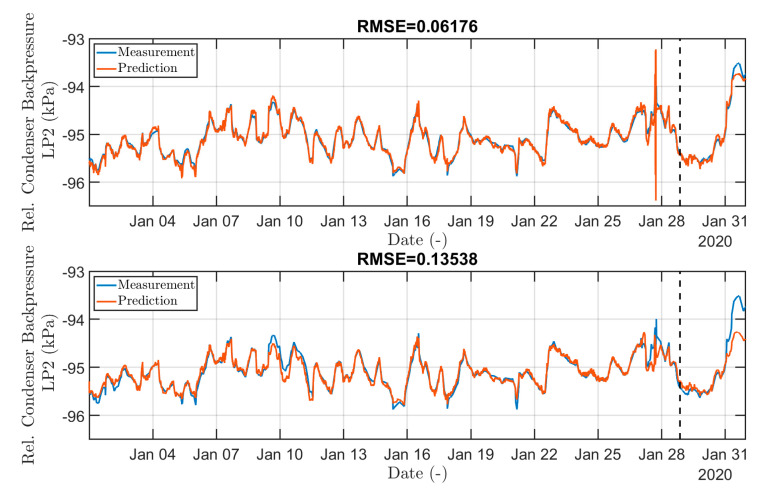
Comparison of measured and predicted values of relative condenser backpressure for TG2-LP2 for FNN (**top**) and LSTM network (**bottom**), with the last 10% of data serving as a test dataset.

**Figure 9 sensors-24-02056-f009:**
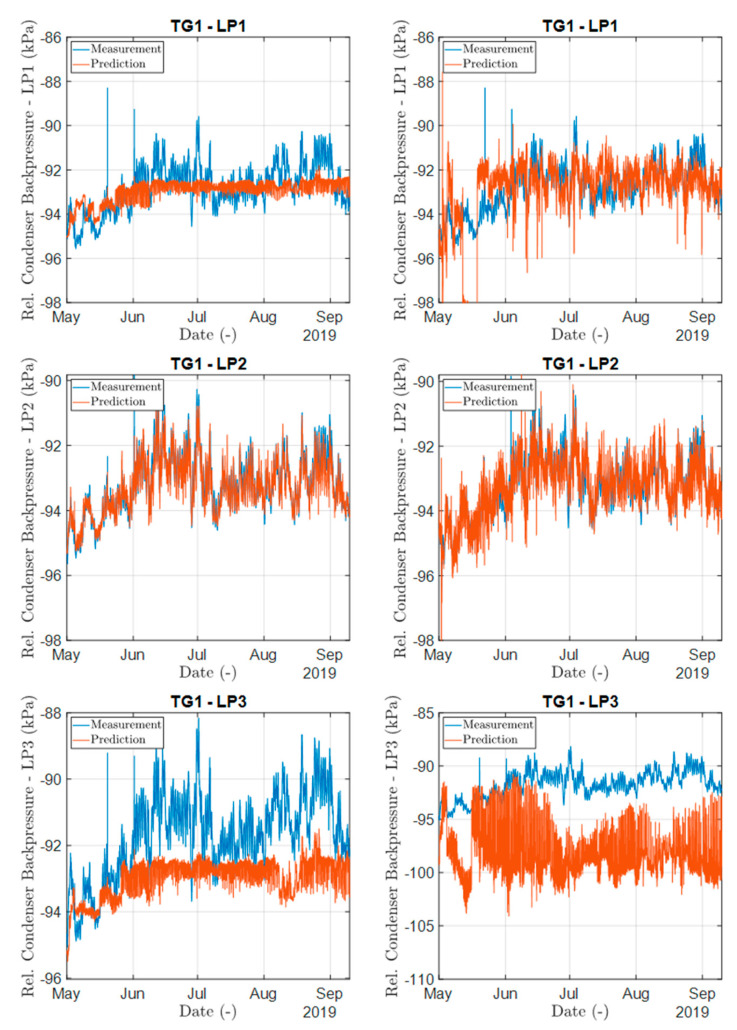
Comparison of measured and predicted values of relative condenser backpressure of TG1, for example, in the period of May 2019–September 2019 for LSTM (**left**) and FNN network (**right**). The training dataset corresponds to part TG1-LP2.

**Figure 10 sensors-24-02056-f010:**
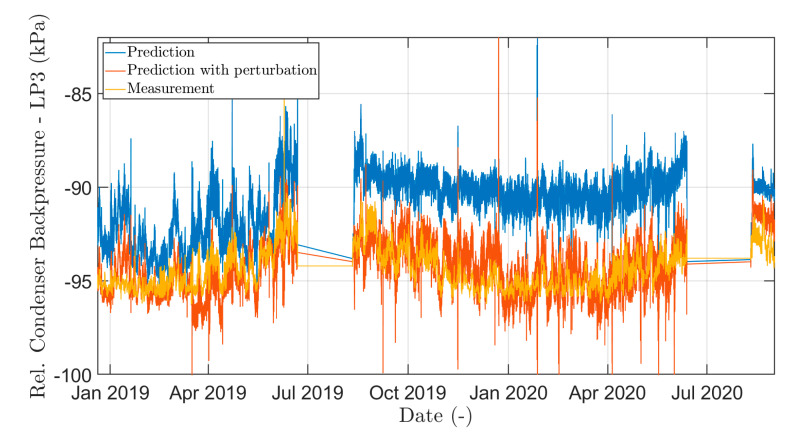
Example of the difference between the predicted values of steam pressure output for LP3 part for the measured input data and when the input data parameter “Steam at 3. extraction from LP3, validated” was increased by 10%.

**Figure 11 sensors-24-02056-f011:**
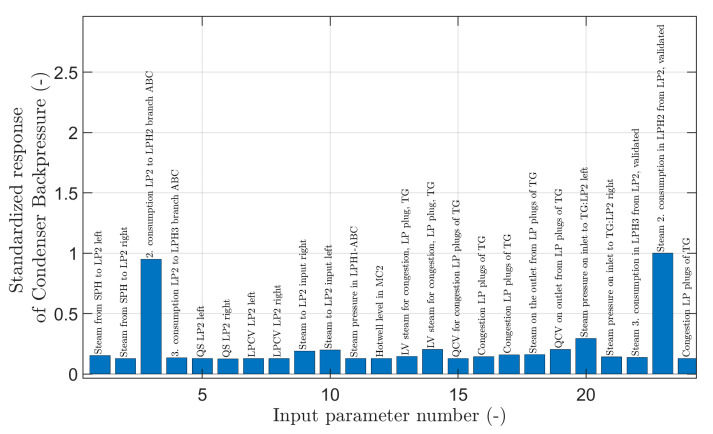
The set of input parameters and the normalized value of their predictive power in the sense of the SHAP method (see Section 4.3).

**Table 1 sensors-24-02056-t001:** Input parameters to the ANN of TG1 LP2; locations of the inputs can be found in Figure 1. The extreme values correspond to the specified power range and the filtering of the input data.

Measured Medium/Part And Its Location	Quantity	Unit	Min	Max
Steam from SPH to LP2—left line	Pressure	kPa	0.3876	0.6327
Steam from SPH to LP2—right line	Pressure	kPa	0.3891	0.6327
LP2 isolating valve—left line	Rel. position	%	80.84	100.03
LP2 isolating valve—right line	Rel. position	%	99.94	100.09
LP2 non-return (stop) valve—left line	Rel. position	%	97.38	99.97
LP2 non-return (stop) valve—right line	Rel. position	%	98.50	100.18
Steam at inlet to LP2—left line	Pressure	kPa	−11.21	1.00
Steam at inlet to LP2—right line	Pressure	kPa	−11.35	1.00
Steam at inlet to LP2—left line	Temperature	°C	242.4	250.9
Steam at inlet to LP2—right line	Temperature	°C	235.29	253.76
Steam at 3. extraction from LP2	Pressure	kPa	59.77	86.62
Steam at 3. extraction from LP2, validated	Pressure	kPa	56.64	84.72
Steam at 2. extraction from LP2, validated	Pressure	kPa	−30.89	−2.03
Steam from LP2 to LPH1	Pressure	kPa	−75.30	−61.83
Steam at injection to LP2 seals	Pressure	kPa	0.7753	0.9687
Steam at injection to LP2 seals	Temperature	°C	24.54	43.62
Isolating valve at injection to LP2 seals	Rel. position	%	−0.067	16.077
Steam in LP2 seals—sensor 1	Pressure	kPa	3.3472	12.896
Steam to LP2 seals—sensor 2	Pressure	kPa	3.3044	12.832
Steam to LP2 seals—virtual sensor	Pressure	kPa	0.6249	14.46
Steam at suction from LP2 seals	Temperature	°C	142.2	148.3
Isolating valve at suction from LP2 seals	Rel. position	%	0.2686	56.18
Hotwell level in main condenser 2	Water Level	cm	69.17	106.64

**Table 2 sensors-24-02056-t002:** Output parameters of the ANN for TG1 LP2. The extreme values correspond to the specified power range and the filtering of the input data.

Measured Medium and Its Location	Quantity	Unit	Min	Max
Steam at the outlet from LP2—rear	Temperature	°C	34.81	49.55
Steam at the outlet from LP2—front	Temperature	°C	34.48	49.40
Steam in main condenser 2	Temperature	°C	28.93	47.02
Backpressure in main condenser 2	Pressure	kPa	−96.49	−89.83

**Table 3 sensors-24-02056-t003:** Hyperparameters of LSTM neural network, tested variants, and selection of final settings. ADAM—adaptive moment estimation; SGDM—stochastic gradient descent with momentum; RMSProp—root-mean-squared propagation.

Hyperparameter	Final Value	Tested Values
Number of hidden units	24	[20, 24, 32, 48, 50, 64, 75, 100, 128, 150, 200]
Solver	ADAM	[ADAM, SGDM, RMSProp]
Maximal number of epochs	200	[50, 100, 125, 150, 200, 500]
Size of minibatch	32	[32, 64, 125, 264]
Length of sections in minibatch	512	[512, 1024, 2048]
Initial learn rate	0.005	[0.001, 0.005, 0.01, 0.02, 0.1]
Learn rate drop period	90	[80, 90,100,125,200]
Dropout layer	Yes	[Yes, No]

**Table 4 sensors-24-02056-t004:** Hyperparameters of FNN, tested variants, and selection of final settings. Trainscg—scaled conjugate gradient backpropagation; Trainlm—Levenberg–Marquardt backpropagation; Trainbr—Bayesian regularization backpropagation; Tansig—hyperbolic tangent sigmoid transfer function.

Hyperparameter	Final Value	Tested Values
Training algorithm	Trainscg	[Trainscg, Trainlm, Trainbr]
Maximal number of epochs	3000	[100, 1000, 3000, 5000]
Number of hidden layers	2	[1, 2, 3]
Number of neurons in the 1st hidden layer	24	[12, 24, 48, 96]
Output activation function	Tansig	[Tansig]
The amount of time delay [minutes]	2	[0, 1, 2, 4]

**Table 5 sensors-24-02056-t005:** Top five input parameters with the largest influence on the output “Backpressure in main condenser 2” (1—largest influence; 0—lowest influence).

Parameter Name (Physical Quantity)	Standardized Response
Steam at 3. extraction from LP2, validated (pressure)	1.000
Steam at 3. extraction from LP2 (pressure)	0.949
Steam at inlet to LP2—left line (pressure)	0.293
Isolating valve at injection to LP2 seals (rel. position)	0.203
Steam at inlet to LP2—left line (temperature)	0.198

## Data Availability

The data presented in this study are available on request from the corresponding author. Note that access to the data may be restricted to certain parties or persons due to requirements by Nuclear law.

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
