# Peer review of "An Investigation of Efficiency Issues in a Low-Pressure Steam Turbine Using Neural Modelling"

_sensors, 2024, doi:10.3390/s24072056_

Round 1

Reviewer 1 Report

Comments and Suggestions for Authors

Author Response

Responses to the review can be found in the attached file.

Reviewer 2 Report

Comments and Suggestions for Authors

The paper presents a very interesting work on the use of neural networks for modeling steam turbines. These models were used to evaluate the operation of the machines and identify the cause of some anomalies.

The paper is fairly fluent to read and presents the methodologies in some sections clearly, in others in a manner that is not very sufficient. 

Below is a list of points that the authors should fix to improve the quality of the content presented: 

1) The abstract does not clearly present the work that has been done. It introduces the problem but dwells little on the technical details and results obtained. It should be rewritten.

2) to the authors' knowledge, their paper is the first to have addressed the problem (NN for steam turbine modelling), but in fact it does not. some of the papers that addresses the issue of modeling steam turbines using neural networks is the following: 

doi: 10.1109/ICAwST.2017.8256435

I may suggest (to my knowledge) the authors of this work were the first to use LSTMs for that purpose.

4) some things should be clarified:

4.1) pg 4, line 137 "The models then selected only 25 parameters..." what does it mean? Elsewhere in the paper I read that experts identified the most significant set of measures

4.2) RNN are not the "opposite" of FNN... I wouldn't use that term

4.3) pg 7, line 254: Absence of? what? I'm sorry, it is not clear to me

4.4) equation 11, pg 7, \phi is not defined. what is it? 

4.5) please better explain equation 12. also define g w_n. what is it?

4.6) pg 8, line 272, define N->N~. What is N~?

4.7) Please define where QS, LPCV and QCV are installed. In which part of the steam circuit? maybe you can add the information on figure 1

4.8) what are the ranges of the variables (min-max) in the table 1 and 2?

4.9) please, also better explain the measures.. some are not clear for me (e.g. LP steam for congestion, LP plug, TG.... etc.). it is a very detailed information, but it is necessary for the reproducibility of the experiment

4.10) pag 9, row 313. Authors say that they removed the samples... how they managed removed samples in the context of timeseries? how you filled the missing informations? 

4.11) authors states that 90% of the dataset was used for training/validation and 10% test. datasets are timeseries. How did you divide the datasets? 

4.12) why validation dataset? did you use early stopping? it is important to explain this point. And eventually.... how you divided the 90% of the dataset for training and validation?

4.13) you often mention HBD.... it is not clear to me the meaning of this. how you used HBD?

4.14) pg 11, first rows.. it is not clear to me how you used HBD

4.15) you mentioned FNN with memory. What does it mean? is it a time delay neural network (TDNN)?

5) why a sampling rate of 1 minute? please explain the motivation, it seems very low to me

6) authors shown several plots, but the accuracy of the models is never described. How is the accuracy of the model for training? for validation? for test? or each model. Please add text about it and also a table for summarizing MSE and other performances you used to evaluate the models

7) it is not fully clear how you exploited the inputs in the FNN and LSTM models. can you better explain how you created the structure of each one? 

8) It is unclear to me what methodology you used in using the models to identify the cause of the anomaly. Can you elaborate on that?

Minor corrections and hints:

1) a table with math simbols and acronyms could be useful

2) random search is better than grid search. Computationally faster and you can find better solution for hyperparameters

Comments on the Quality of English Language

The paper is closer to a report rather than a scientific paper. it is a stylistic choice that I sometimes appreciate, sometimes not. The quality of grammar can be improved, as can the use of certain terminologies. Revise the text a little bit, but overall it is quite fluent

Author Response

(The authors gave the same response as above.)

Round 2

Reviewer 1 Report

Comments and Suggestions for Authors

The authors addressed all of my questions and concerns. I recommend the publication of this manuscript.

Reviewer 2 Report

Comments and Suggestions for Authors

No additional improvements needed, the quality of the article is now good enough to be published